# Dropping Out or Continuing Playing—A Case Study of Adolescent’s Motives for Participation in Football

**DOI:** 10.3390/sports11070128

**Published:** 2023-07-03

**Authors:** Jostein Bergin, Pål Lagestad

**Affiliations:** Department of Teacher Education and Art, Nord University, 7600 Levanger, Norway

**Keywords:** drop out, participation, sport, football, adolescents

## Abstract

The purpose of this study was to investigate dropout and continuation motives among boys in youth football. Semi-structured interviews were conducted with all players from a former football team, consisting of 13 young people who were 17 years old when the in-depth interviews were conducted. Short interviews with the same players five years earlier (at the age of 12) were also included as part of the data. This strategy was used to gain a longitudinal perspective and a better insight into the participants’ experience of organized football. Including in-depth data from all players at a football team—both players that dropped out and continued playing football, and also including some short interviews data from a longitudinal perspective, this study bring new findings into the discussion about sport participation. When the in-depth interviews were conducted, six of the participants had dropped out of football, while seven were still active. The results show that social factors, ambition, other interests and differences in skill and physical development, were all reasons for dropout from this team. Among those who continued playing football, the social aspect of football, their love for the sport, competitive instinct and the training benefits the sport provided, were reasons they continued playing. Furthermore, the findings showed that those who reported the lowest ambitions at the age of twelve, had dropped out of football, while the one with the highest ambitions had continued playing football. Our findings indicate that ambition can be a crucial factor in relation to participation in football, and also that the social aspect of football is an important motivational factor that coaches and parents should be aware of.

## 1. Introduction

According to the strategic focus areas of the Norwegian Sports Confederation (NIF), sport should stimulate lifelong enjoyment of sport so that children, young people and adults continue to be recruited [1]. It should also ensure that as many people as possible continue for as long as possible, regardless of age, ambition, or level of skill [1]. Nonetheless, the proportion of adolescents who remain active in Norwegian organised sport, reduces markedly during adolescents in Norway [2], as in countries all over the world [3]. The survey “youth data” from 2016 to 2018 indicates that 60% of those participating in youth sport, were no longer a member of a sports team by the time they reached the age of 17–18 [4]. 

### 1.1. Sport Participation Benefits

Sport is an important socialisation arena for youngsters, where they gain experience of relationships and are confronted with emotionally challenging situations [2]. Physical activity and games are highly valued by children and young people the world over and are therefore a good means of developing social skills [5]. In the future, the importance of retaining young people in sport will increase still further, as children and young people in Norway, and the rest of the world, engage ever increasingly in sedentary activities [2,6]. Research has shown that the amount of inactive time among young people increased significantly between 2005 and 2011, and that many young people today fail to meet the recommended 60 min daily of physical activity [6]. A substantial dropout from sport can have major consequences for public health, as regular physical activity is very important for normal growth and the development of functional qualities such as motor skills, as well as psychological and social factors [6,7]. It has been shown that participation in physical activity and sport is associated with a lower incidence of psychosocial health problems [6], and that young people who participate in sport have better “cardiovascular fitness” than young people not playing sport [8]. Young people who give up sport also differ negatively from those who continue in terms of quality of life, risk-taking behaviour, and participation in other leisure activities [4]. Because of the importance of sport as a social arena and an arena for physical activity among adolescents, this study will look more closely at why young people stop playing football, as well as what leads them to continue. 

### 1.2. Previous Research

Previous research indicates that several factors are associated with participation in organized sport. Studies focusing on the reasons for dropping out have concluded that these are complex, and that there are usually several in play simultaneously [9,10,11,12,13,14]. According to a recent study, 18% of the dropout from youth sport can be linked to high demands from the sports teams [13]. In this study, it is proposed that this is a consequence of sports becoming too serious during adolescence, the demands placed on the athletes increasing in line with the other demands of everyday life. This is supported by other studies pointing to school as another arena in which there are increasing demands [2,11].

Several studies have pointed to a lack of technical skill [15,16] and of certain physical characteristics, as height and fitness (VO_2_-max) as leading to dropout [8,16,17,18]. It is also found that many lose the motivation to pursue their sport during their youth, because they realise that they are not going to become top athletes later in life [18]. This may also arise from them no longer experiencing themselves mastering the sport as well as their peers. Also, ideal body discourses, performative body discourses and a lack of physical competence, is found to affect adolescents’ decisions to engage in or drop out of sport [18].

There are also studies pointing to so-called structural limitations such as, for example, time [9,11,14,19,20,21]. Among other issues, travelling to matches or too many training sessions in a week can lead to sport taking up too much of everyday life. This is also found in a study Persson et al. [11], who identified three main reasons for opting out of sports; lack of enjoyment, time, school pressure, and that the sport became too competitive and serious. Owen et al. [21] also found lack of enjoyment to be associated with drop out from sport. The impossibility of reconciling sports with school has also been found as a dropout factor in other research [19,20]. Another reason for the time available being insufficient may be young people wanting to prioritise other activities [10]. This can be, for example, wanting to spend more time with friends, or meet a boy-/girlfriend [13]. Injury is also pointed to as a reason for dropping out [13]. It is suggested that dropout because of injury may result from low motivation to work to get healthy again, or that the club and the coaches follow up the player only poorly during the period of being injured [13]. It is also found that some trainers are unable to relate to young people, and that this lack of competence/understanding leads to young people withdrawing [13]. In certain cases, it can also be that players experience themselves as being badly treated by their trainer, some giving this as a direct reason for their decision to stop playing [12]. An intervention study during a youth football program by found that increasing tactical-technical competence, successful game performance and player autonomy, improved variables related to dropout, and prevented dropout [22].

Several studies showed that participation and drop out from sport is related to social relationship [10,12,13,15,20,23]. In his study, Flæthe [9] found that dropout was 7.8% higher among young people weakly connected to their friends, than among those with an ordinary relation to theirs. This agrees with a study by Jakobsson et al. [11], who found that a principal reason for young people engaging in sport is exactly because of the feeling of belonging to a group. Back et al. [15] found that social support was negatively associated with drop out. Also, Gatouillat et al. [20] pointed to friendship as a reason for dropout in their study. Abadi and Gill [23] found that social support was greater for girls who continued participation in sport, than for those who dropped out. If the participant feels a lack of this social belonging, it may lead to social dissatisfaction, which may, in turn, lead to eventually becoming tired of doing sport [13]. Young people feel under pressure to meet the needs of both family and friends and, if this pressure becomes too great, this can lead to them dropping out [9]. Jaf et al. [24] found that youth who disclose their whereabouts to parents and whose parents practice control are less likely to engage in delinquent behaviors, and, in turn, more likely to engage in organized sports.

The theoretical framework of the study is related to Deci and Ryan [25] self-determination theory. Deci and Ryan point out three basic needs in all humans: autonomy, relatedness, and competence. These factors are essential for optimal motivation, integration, wellness, and well-being, which in turn lead to intrinsic motivation. Intrinsic motivation is about doing an activity because it is interesting and provides its own reward by satisfying one’s basic needs for autonomy, competence, and relatedness [25]. It is the prototype of self-determination in their self- determination theory. Self-determination theory emphasizes the importance of the social environment of students for personal growth. According to participation in football, we will argue that relatedness may be of great importance. Relatedness refers to feeling connected to others, to caring for and being cared for by those others, to having a sense of belongingness both with other individuals and with one‘s community [26]. 

Against the background of the above discussion, the study examines the following research question: “Which factors influence boys playing youth football to either stop, or continue playing organised football?” In relation to the previous discussion, it was hypothesized that the findings would be related to social relationship, structural limitations, and technical skills.

## 2. Materials and Methods

To answer the research question, a case study design with both in-depth interviews and short interviews were used [27]. The research project was approved by the Norwegian Centre for Research Data, fulfilling the ethical standards for empirical research. The respondents were fully informed about the protocol prior to participating in the study, and written consent was obtained from all. The study was conducted in accordance with the Declaration of Helsinki.

### 2.1. Participants

The authors were aware of a football team that had played together throughout much of their childhood, but which had now fallen apart when some of the players had been offered membership of other clubs, and other players had stopped played organised football. All players from the team (n = 13) agreed to the use of data of their ambitions in football, taken by their coach when they were 12 years old. All players also agreed to be in-depth interviewed five years later, at the age of 17. At the age of 12, the team played at a local level as all other team in the area and was rated as one of the best teams within their local area (within 50 kilometres). At the age of 13–15, the team played at the first level in their region (within 150 kilometres) and was rated as the 10–15 within that region. At the age of 16–17 the team played at the second level and was rated between 15–20 in their region. The team had two different main coaches during the period, and from the age of 14 there was two assistant coaches at the team. The team got medals at an international tournament at the age of 14 and 15. At the age of 12, the team consisted of 70% of all the boys in their region, and at that time they played much football at leisure time and during recesses at school. 

The participants came from a local community with three different primary schools within the same region, but at the age of 13, they all were included in the same lower secondary school, which is normal in many Norwegian regions. The participants came from different social class, where most of the parents had a medium income and education level, but also a few parents with low or high income and education level, according to Norwegian standards. In this way, the participants probably reflect the natural variations within social class in Norway. At the age of 12 the team had organised football training twice a week, and three times a week from the age of 13. The coaches reported that most of the players in the team were relatively ambitious, but also that some of them were participating out of social reasons. 

### 2.2. Procedures

During the interviews when the participants were 12 years old, a semi-structured interview format [27] was used. An inductive design with open questions were chosen regarding their ambitions and aims for football. The answers were written down by hand in a notebook and lasted between 10 and 15 min. Semi-structured interviews [27] were again used for the in-depth interviews when the boys were 17. Open questions were chosen for the interviews, but also questions in which the participants were questioned about factors shown by earlier research to have a bearing on participation in football. The questions used were: “Have you stopped or continued playing football?” “Can you say a bit about why you decided to stop/continue playing football?”, “Do you miss playing football/why do you still play football?”, “Can you say a little about the way your ambitions have influenced your participation in football?”, “which factors do you think were important for your decision to stop/continue playing football?”, “Have injuries affected your participation in football?”, “Have the coach affected your participation in football?”, “Have social relations within the team affected your participation in football?”, “Did your social relations to players at the team affect your participation in football?”, “Did the ambition of the team affect your participation in football”. Emphasis was given to following up the answers with further questions where this seemed natural, and many of the open questions did require many follow-up questions. The in-depth interviews were conducted wherever the participant wanted it to be, in order that they feel as comfortable as possible during their interview. Some were held at the school they attended, the others at the participant’s home or in hired locales in their home area. The interviews were audio recorded and lasted between 30 and 45 min.

### 2.3. Analysis

In the analysis phase, the notes from the first interviews and the audio recordings from the later interviews were entered into the qualitative analysis program NVivo 12 Plus. The interviews were transcribed verbatim, and coded so that the participants were anonymised and given fictitious names. After the interviews were transcribed into NVivo, their content was analysed, and the meaning condensed in that quotations and short paragraphs were reduced to shorter and more concise formulations [28]. The analyses were based on a case study approach that included all the participants’ answers on the questions collected during the interviews, in which the participants’ experiences related to football were taken as subjectively true [29]. With such an approach, we did not ask the coaches about their experiences or perceptions. The data were based on subjective constructions, which the adolescents constructed as part of their own interpretations and reflections on what has occurred in their football team. The analyses were based on transcribed answers focusing on meanings, as described by Johannessen, Tufte, and Kristoffersen [27]. Opinions and statements were identified according to the theme of dropping out or continuing with football, then concentrated, condensed, coded, and categorized in units of analysis [30]. In this process, the participants’ statements are assigned codes that are classified into categories [31]. The data are sorted based on these categories to elucidate patterns, similarities, relationships, or differences between the statements. The analysis and the interpretation also followed hermeneutical principles, in that the interpretation process led to an increasingly deeper understanding of the statements in the interviews according to our comprehension of the whole and the parts, and to understandings that were free of contradictions or logical flaws [32]. 

The transcribed text was read several times. Reading the text led to the creation of categories from the participants statements related to distinct reasons for dropping out from or continuing with sport. First, all responses that referred to dropping out from football were marked and coded as belonging to the category to be ‘drop out’. Subsequently, all text that was coded ‘drop out’ was read, and four subcategories were created. These categories were: The experience of a poorer social relationship, lack of ambition, growth of other interests and re-prioritising of time and physical condition. Furthermore, all the text labelled ‘continuing football’ was read through in order to extract the categories that seemed to be related to the phenomenon of continuing football, by condensing the meaning of the statements. Four subcategories that dealt with the phenomenon ‘continuing football’ were created. These categories were: The social aspect of football, getting fit through football, finding it fun to play and compete, and ambition as a driving force. After, various alternatives for interpretation and perspectives were discussed among the authors. This contributed to intersubjective consensus in the analysis and strengthened the credibility of the findings. A narrative structuring was then carried out under each of these categories [28]. In this, the participants’ statements were placed in context with each other, creating a story with a common thread. Six of the players from the original team at the age of 12 had dropped out of football, while seven continued playing football. The categories from the analyses are presented in Table 1, where the quotations used in the results section are structured according to the analytic themes. The participants have been given pseudonyms in the presentation of the analysis. 

## 3. Results

### 3.1. Reasons for Dropping Out of Football 

It seemed appropriate to structure the results in two categories: reasons for dropping out among those no longer playing football, and reasons for continued participation in football. The analysis showed that there were four main reasons leading to the participants deciding to stop playing football: the experience of a poorer social relationship, lack of ambition, growth of other interests and re-prioritising of time, together with lack of physique and skills. These are further discussed below.

#### 3.1.1. The Experience of Poorer Social Relationship 

All those who had stopped playing football pointed to the social aspect of football as being decisive. The analyses suggest that this social aspect of participation in football is highly important. Several of the participants gave the social aspect as a significant reason for them having started to play football. The interview data also suggest that this social aspect of football can be divided into two subcategories—the social within football and the social outside football. Within football, this includes team togetherness, and that the participants feel part of a community. The social aspect outside football concerns some people having friends outside of sport involved in other things who may help to lure participants away from their sport.

A good example of the importance of the social inside football comes from Jim. He tells us that he gave up football around a year ago. He claimed that the main reason for him stopping playing was that the social element within the team became worse and worse. He found that a bad atmosphere developed among the group of players, and then he simply got tired of showing up for training. Simon supports him, also pointed out that the unity of the team became worse in the time before the team was disbanded. Simon also says that some new players joined the team, and this contributed to the creation of a bad atmosphere. He felt that the team split up into smaller groups, and that the different groups did not get along with each other at all well. Simon’s story agrees with that of Joe. Joe was one of those who said that he played football because his friends did, and because they had so much fun together when they were younger. In the end, when the team started to be more ambitious, he chose to quit, because he felt that a good part of the social aspect had been lost. When asked if he missed football, he replied: “I don’t miss football as such, but I do miss having somewhere to go and a group to train with”.

Didric points to the transition from secondary school to upper secondary involving making a lot of new friends, as he had done. These new friends often belong to other teams or were not involved in organised sport at all. 

James had the same experience as Didric: “It was miserable when my friends met up and I couldn’t because of the football”. Tom had not given up football completely, but when interviewed he had not been training for several months because his friends had all either given up or joined other clubs. When asked why he had not been training, he replied with a sigh: “I’ve heard that my mates have gone to other clubs or given up, so I’ve less interest in going to training. None, really”. James’s story exemplifies the importance the social aspect has for continued participation in football. 

#### 3.1.2. Lack of Ambitions 

All those who had dropped out of football, pointed to lack of ambition as a reason for dropping out. Some said that they had lost their ambition and therefore got bored with it, whilst others said that the gap between their and the team’s/fellow players’ ambition eventually became too wide. Tom was a player who really wanted to give it a go when he was younger. Recently, however, he had found that he no longer had the same spark. He told that he no longer had the aim of becoming a professional footballer, and that he therefore lacks the ambition to make him meet up for training: “I’m never going to be a footballer, so why not just lie in bed?” A good example of the gap in ambition between the players on a team leading to dropping out, is Didric. When he was asked if the ambitions among the other players on the team influenced him somewhat, he answered: “Part of the team was up at the pitch all the time and trained and played because they wanted to be best, and when I play just for the fun of it, there’s a bit of a difference there”.

Joe also stated: “When the team started to be at the top and everything became more intense, I felt that this wasn’t going to be as much fun, since the only thing was to win”. Simon felt that one of the reasons why the team was disbanded, and that some people quit, was due to these differences in ambition: “There were some who wanted football as just a game, and those who wanted to put more into it, and that’s how it went”.

The analysis of the interviews conducted by the coach when the players were twelve years old points to an interesting finding. Of the six who have now stopped playing football, there was only one who described having big ambitions as a 12-year-old, while the great majority of those still playing were ambitious regarding football when they were twelve. Those who had no ambition said that they played football for the fun of it, to keep fit and because it was social. Dropping out in this way may then be due to natural causes such as a lack of an inner drive or interest in the sport. This partly came out through the interviews, where several of those who had given up, said that they had no particular interest in football apart from that they played themselves: “I knew that it wasn’t really what I wanted to do, but to have something to do, so I kept it going for years”. He said that he had nothing against football and liked to play with friends in his free time, in summer for instance. It was just, in the end, that organised football hadn’t gone the way he had wanted.

Another interesting finding is that one of the players who had given up football, said as a 12-year-old that he had great ambitions of becoming a professional footballer. In the depth interview five years later he said, however, that he had never had any wish to be a professional footballer. This ambivalence in the two interviews, five years apart, is interesting.

#### 3.1.3. Growth of Other Interests and Re-Prioritising of Time 

Of the six who had given up football, four said that football took too much of their time, or that they wanted to use their time differently now. One of the reasons that football came to take too much time was that they travelled further to games, as James pointed out. He said that he thought it was fun to train, but he felt that travelling to play matches became too demanding. He found that the away games were too far away, leaving him with no time to do other things. He also said that he had had a trial period during which he trained but didn’t play in the games, but he gradually came to feel that even this became a struggle, as he preferred to spend his time on other things. Examples of things he now wanted to do included being with friends, working out at the gym and going to parties. To emphasise this, he said: “I think it was good to be able to decide for myself when I had to come up with something to do”. 

Jim said that starting at upper secondary school made it more difficult to prioritise football. He was in a vocational set at school, and therefore had several periods as an apprentice in different firms. He felt that this didn’t combine well with football, and he wanted to prioritise completing these apprenticeships ahead of football training. He also said that he did not think that he wanted to return to football again later, as he had chosen a more career-based path now and thought this was better than “wasting his life on football”, when he didn’t really know what he wanted from it. Didric also wanted to deprioritise football. When he began at upper secondary school, he was in a different set to the others, and he found that he developed somewhat different interests. Amongst other things he began with gaming and found that the interests of the others in the team didn’t go in the same direction.

Simon, who stopped playing football, didn’t however give up football entirely. He started to referee matches instead of playing himself. He also started working part time and felt that the total load of everything eventually became too much, and that he simply did not have time for everything he wanted to do: “It all became a bit too much when I was working, refereeing as well as playing”.

#### 3.1.4. Lack of Physique and Skills

Two of those who stopped playing football gave, respectively, lack of physique and lack of skill as being decisive in their giving up. Didric pointed to his own physical development as being one of the main reasons for him giving up, arguing that the main reason for him stopping was that he could no longer compete with the others in his age group. He found that everyone around him became bigger, stronger, and faster, and that left him no longer taking pleasure in playing: “I came to puberty really late, and when all the others in the team and in the opposition got much bigger than me, and I became just little and kiddy in comparison”. Didric said that he thought it was a lot of fun playing football at times, but when he was no longer able to compete with those of the same age, he lost some of his motivation: “In the end, I couldn’t compete, because they were all bigger, stronger, and speedier. I remember it being very frustrating”.

For Joe it was not so much the physicality, but rather that the level of football skills eventually became too high for him. He says that he was always aware that he was not the one who spent the most time on the pitch and that he was not the best player, but eventually he simply found that there was too much of a difference.

### 3.2. Reasons for Continued Participation in Football

Of the seven who continued to play football after the team had been dissolved a year earlier, one played for the senior team of the old club. Two played for development teams for juniors in the municipality, and four played in open-access clubs for juniors in the municipality. The analysis indicated that there were four main reasons why the participants continued to play football: The social aspect of football, getting fit through football, finding it fun to play and compete, and ambition as a driving force. These findings will be discussed below.

#### 3.2.1. The Social Aspect of Football

Among those still actively engaged in football, its social aspect was put forward as being important for their participation. Most, in fact, gave this as the main reason for them starting in football at all. All pointed to the importance of having friends within the sport, as this makes it much easier to get into it yourself, especially at an early age.

One of those pointing to the importance of the social aspect was Gary, who had gone over to the new club in the municipality. For Gary, the social aspect was particularly important reason to participate in football: “I would have liked it to be like it was at the age of, when all my friends were together”. Gary also stressed the importance of enjoying yourself with those you are in a team with. He says that the hard training feels much easier when he is with his friends: “There’s a big difference between running intervals with people you enjoy being with, and with people you don’t enjoy being with”.

Harry also says that this social aspect has played a large part in his participation in football. After the junior team he played for was dissolved, he tried the senior team of the same club, but quickly came to feel that it wasn’t the same. Most of them were older than him and he didn’t know them in the same way as those in his own age-group. He therefore decided to go over to the new team in the municipality where he had the chance to both play football with some of his friends, as well as being able to do other things at the weekends and on the days without training.

Those playing in the more elite team also pointed to the social aspect of it as being important. For a long time, Kevin played in the old club with boys who were a year older, but switch to the elite team. He says that he felt confident about this choice as he went to class with some of those who played there, and finally got to play with people of his own age.

Jamie also pointed to how the social aspect had affected him until he made it to the higher-level team. He was in a class at school with many others who played football, and, later, there was a little group of two or three who played together all the way through secondary school. He said that, earlier, it was important to him that his friends also played football, but that more recently he had been so preoccupied with improving and therefore chose to move on to the elite club. He found that here there was less nonsense at training, and he had better opportunities to improve his football.

Marcus, who still plays football, understands the social aspect is important for continued participation in football. He pointed to participation in a particular summer cup competition as having been the biggest social event for the team each year. When asked if its cancellation had affected him especially, he answered: “You could say that I lost a certain amount of motivation. That was the big carrot every season, and then it was taken away”. He thought that it was the cup competition they used to be part of in the summer that motivated some of them to show up, and that attendance only began to drop off when it was finally cancelled. 

#### 3.2.2. Getting Fit through Football

Everyone who participated in organised football agreed that one of the reasons they played was that this was a way of getting fit. Several said that football was thought of as a kind of “free” training, because it is so pleasurable that they do not find it stressful to train in this way. Marcus gave as one of the reasons he went to football training that it gave him the feeling of having got fit. He said that he sometimes went to the gym and other places to get this feeling of fitness, but not often. His preference was to get it through football training as he thought this the most pleasurable way of doing so. Tom said that his great ambition to play football had dwindled through time, so now it was mostly a question of keeping fit. He was clear that he believed that everyone must find the best way of getting fit, and, for him, this was playing football: “It’s fun playing football. Free fitness training!”

#### 3.2.3. Finding It Fun to Play and Compete

All those who had continued playing football said that they did so because it was ‘fun’, and that it was the activity they were most tempted to pursue. When Chris was asked why he still played football he answered that “love of football” was the reason. He went on to say that he had been troubled by injury from time to time but had never lost his desire and had never seen stopping as being an alternative. He said that, at times, he was so consumed by it that he almost forgot both school and private life, and pointed out: “I have a winning mentality, so I just have to win at something”. The importance of competition in football in relation to keeping playing was supported by several of the others. Harry also said that he likes to compete, and that it a reason for him to play football. Jamie put forward the instinct to compete as one of the main reasons he plays. In his interview, Jamie said that he continues to work at his football, wanting to see how far he can go with it.

#### 3.2.4. Ambition as a Driving Force

Analysis of the short interviews carried out by the trainer when the boys were 12-years-old, points to an interesting finding. Here, six of the seven players who still play football point out that their ambition with football was to become a professional football player, in teams like Liverpool and Manchester United. That so many with ambition had persisted in playing football, may point to ambition being an important driving force towards continuing to play and not dropping out, however with lower ambitions. This was the case with Chris. In the in-depth interviews he claimed: “For a period, I was so into football that I forgot everything else. I changed teams because I wanted to train as much as possible and become as good as possible”. Of those six who expressed the great ambitions as footballers when they were interviewed when they were 12, there were, however, only two who still dreamed of playing professionally when interviewed as 17-year-olds. This means that 4 players who had once had ambitions to be professional, no longer have them. Analysis of the interviews suggests that this change may be the result of injuries, together with other interests having meant that their investment in football wound down, but also because of a reality check. Gary stated: “I probably won’t go up to top leagues in the future, but I still have the goal of playing on a decent senior team when I get older. I want to have fun with football and go as far as possible”.A couple of the players also found that reality caught up with them and that they gradually began to realise that it might be quite difficult to become a professional footballer, as Patric: “I trained a lot more before, when I had ambitions to play in the highest national league. Those goals have changed a bit now that I am older and more realistic”. However, due to the use of short interviews at the first interview, in-depth descriptions of their ambitions were not collected at this time, and careful considerations should be given to this finding.

## 4. Discussion

The aim of the study was to examine factors that influenced boys to either stop or continue playing organised football, and the discussion is structured in relation to the central categories in the study; reasons for dropping out among those who stopped playing football, and reasons for continued football participation.

### 4.1. Reasons for Dropping Out among Those Who Gave Up Football

As Persson et al. [11], our findings also illuminate how several complex factors, such as interpersonal relationships, structures and priorities, contributes to different senses of belonging among youth, and thereby dropout or continuing with football. The results point to one of the main reasons for dropping out of football being changes to the social environment. The finding is in accordance with the self-determination theory [25,26], pointing to relatedness as one out three basic needs in all humans that are essential for optimal motivation, integration, wellness, and well-being.

The importance of the social environment according to sport participation is supported by previous research [10,12,13,15,20,23]. Torp [13] pointed to lack of social belonging as leading to drop out, and Jakobsson et al. [12] highlight belonging to a group as one of the reasons why young people engage in sport. Several of the players felt that participating in football was difficult to combine with maintaining a social network outside football, and therefore chose to quit so they had the opportunity to participate in social circles outside football. Friends going over to other clubs or stopping playing football were other reasons for drop out given by our informants. This lack of belonging to the group has also been pointed to as being important in another study [12]. In this study, it emerged that several found that the social climate in the original team deteriorated over time. The finding of lack of belonging is supported by an earlier study finding the dropout rate among young people with weaker relationships with friends to be 7.8% higher than among those with average relationships with friends [10]. 

Our findings points to the importance of ambition at a young age as a driving force for participation in football, as Enoksen [33] also found. Lack of ambition can lead to dropping out of football, as the results from our study suggest. Our findings suggests that large differences in ambition can lead to drop out. If more than half of the team want to progress, this can lead to group pressure on those who don’t want that. The players then feel pressured to satisfy their friends’ needs, and if this pressure becomes too much, it can lead to them choosing to drop out [34].

The results show that several of the boys began to experience football as being overly time consuming. Several earlier studies also point to structural limitations, such as time commitment leading to dropping out [10,13]. Together with the other demands of everyday life [13], this may lead to the total load eventually becoming too much for them. Several of the participants pointed out that changes to their schooldays have played a part, something that other studies also have found [11,19,20]. Our findings show that the experience of having fun is important, and Persson et al. [11] referring to having fun and with friends and peers while engaging in joyful activities, as the most often reported reasons for taking part in sports. According to Persson [11], youth sport should be arranged in such a way that the young people are also able to participate in other activities outside sport. 

This study points to the development of other interests, and to the youths wish to make different priorities as reasons for dropping out of football. This finding is supported by Persson’s study [11], pointed out that one of the reasons for dropping out of football is simply that young people want to prioritise other things.

Different physical resources and abilities also appear as grounds for dropping out of football. This finding is supported by earlier research pointing to physical resources as a reason for dropping out [6,8,14,17]. Lagestad [8] refer to a lower oxygen absorption level and height as reasons for dropping out of organised sport. Other studies found that the dropout rate is higher among young people who have come less far in their physical development [16,17,33,35]. We will argue that this finding is related to Deci and Ryan [25] self-determination theory, pointing to the feeling of competence as a basic need in all humans, essential for optimal motivation and well-being. This suggest that young people’s physical starting point must be taken into account when organising sport. As suggested by Vella et al. [36], a solution-oriented approach to dropout from sport is recommended. In light of our findings, we agree with Owen et al. [21], pointing towards the importance of developing sport to be enjoyable, flexible and modifiable to all adolescents’ needs. We also agree with Back et al. [15], who point to the importance of developing a high-quality motivation climate that facilitates motivation and enjoyment. With such a strategy, we point to the study of Fabra et al. [37], who found that autonomy-supportive coaching increased the players autonomous motivation, which negatively predicted dropout intentions. This finding and suggestion is in line with another study of Back et al. [38].

### 4.2. Reasons for Continuing in Football 

In order to prevent drop-out from football, it makes sense to learn from what emerges among those who still participate in the sport. As shown in the results section, the experience of a good and inclusive social environment is seen as essential when the participants are highlighting the reasons they continue to play football. This agrees with previous research suggesting that if coaches master the task of creating a socially supportive environment, this may lead to more people wanting to play sport for longer [12,13]. We also agree with the suggestion by Persson et al. [11], because also our findings illustrate the need to see inclusion in sport in relation to processes in other arenas that are important in adolescents’ everyday life. We find their suggestion that the concept of belonging accentuates some of the meaning related to the processes involved in staying in or opting out of sports. Our findings also support their reflections that the high rate of dropout from sport may indicate that many adolescents experience sports as an arena that becomes less important for a sense of belonging and social wellness as they grow older, because participation in sport comes into conflict with other parts of their lives.

Torp [13] claims that the absence of social belonging can lead to dropout, but in our study, it also led to the players changing clubs.

A wish to compete was also identified as an important reason for continuing to play football. Jakobsson et al. [12] also found that young people reported that they participated in sport because it them the opportunity to compete. Also, the findings of Lagestad and Særensen [39] support the importance of enjoyment in sporting competition, according to well-being in sport. The authors showed that enjoyment in sports competition, is a main predictor of adolescents’ enjoyment of sport. Other research indicates that a high level of sport enjoyment is important for reaching an elite level in sport [40].

Another key finding among those who continue to play football is that they say they use football as a means of “free training” and are therefore aware of the importance of staying active. Given the research pointing to young people increasingly engaging in more and more sedentary activities [2,6], it may also be important that young people are made aware of the value of physical activity, and the many benefits of physical exercise.

Being ambitious appears to influence whether one drops out of, or continues to play, football. The analysis showed the most of those who said they harboured the ambition of being a professional footballer, no longer had this ambition. This appears to be the result of injury, changed priorities, or perhaps most that they began to realise that it would be difficult to fulfil their dream of playing professionally. A study showed that only one in 777 boys in Norway manage to attain this dream [41], and it is natural that ambitions are lowered when you see that the best in your area are starting to be picked up by the top clubs in the country, and you are not. It is reasonable to argue that many young players all over the world have ambitions to play in teams like Liverpool and Manchester United, but that very few fulfil this dream.

## 5. Strengths and Weaknesses of the Study

The results in this study are obtained from short interviews with members of a football team consisting of 13 players when they were 12 years old, as well as in-depth interviews with the same players when they were 17 years old. That all the players from a team were interviewed across a 5-year interval, provides insight into how the individuals in the team were affected longitudinally through time, even if on short interviews were used at one time. The fact that the study includes all players from a team—both the perspectives of those who drop out and those who continue with football, is fruitful.

However, the present study possesses several limitations. One of the weaknesses of the study is that it is not possible to generalise the findings to all youth population from a small non-randomised sample. The team selection is based upon a stratified selection and is biased with such a strategy. Furthermore, the study has not an experimental setting with a control group. However, the results may be valid for Norwegian adolescents, and for adolescents in other countries similar to Norway in relation to the organization of football. Another weakness is that the interviews at the age of 12 were rather short. Using in depth interviews (as at the age of 17) would have been preferred. 

## 6. Conclusions

This study is the first to examine data from all players at a football team, including both players that stopped and continued playing football. Even if the study could be criticized by using only short interviews at one time (at the age of 12) and only in-depth interviews at another time (17 years of age), the study bringing new and somehow longitudinal perspectives into the discussion. 

The results of this study point to four main reasons why the participants chose to leave football: the experience of a poorer social environment, lack of ambition, development of other interests and prioritisation of time, as well as a lack of physical fitness and skills. Similarly, four main reasons for the participants choosing to continue playing football were pointed to: the social aspect of football, getting fit through football, the experience of it being fun to play and compete, as well as ambition as a driving force. 

In agreement with previous studies, this study points in the direction that dropping out of, and continued participation in, football result from several complex mechanisms. Our findings indicate that the social aspect of sport seems to be a highly important motivator. For this reason, it is important that coaches in organised sports are aware of this. Such an approach is also in line with NIF’s guidelines that as many as possible be retained, as long as possible [1]. This includes, among other things, providing social events with the team outside football, as well as participating in cups and tournaments outside the local area as this can contribute to stronger cohesion and stronger group affiliation. There are also things indicating that lack of ambition impacts dropping out, which means that regular conversations with the players are important as this gives the coach an overview of what the players want. It would also seem to be important to have a football offering for those wanting to play football just for fun or because of its social aspect. Not all teams should be ambitious teams. 

This study was carried with a single team that stayed together for many years. This type of situation results in a strong micro-system (families, coaches, peers, significant others, material, social class, and demographic factors) that arguably exerted a strong influence on the participants. Further research should look more closely at how this micro-system influence participation in football, but also how the development of ambition affects the motivation of the participants, and whether a change in ambition is a predictor of dropping out of football using a quantitative approach (questionnaire), with a significantly higher number of respondents. 

## Figures and Tables

**Table 1 sports-11-00128-t001:** Description of the categories, themes and quotations that were derived from the analyses.

Category	Themes	Sample Quotations
**Reasons for dropping out of football**	Poorer social relationship	**Tom**: “I’ve heard that my mates have gone to other clubs or given up, so I’ve less interest in going to training. None at all, really.” **Didric**: “My friends outside were people I wanted to devote more and more time to. When there was something to do in our spare time there was a bit too much of—No, I can’t, I’ve got football training.” **James**: “It was miserable when my friends met up and I couldn’t because of the football.” **Simon**: “There was a lot of throwing of comments back and forth, which caused a lot of irritation over time for me. The main reason [for me quitting] was that there was a bad atmosphere in the team and a lot of comments over time.”
	Lack of ambition	**Tom**: “I’m never going to be a footballer, so why not just lie in bed?” **Didric**: “Part of the team was up at the pitch all the time and trained and played because they wanted to be best, and when I play just for the fun of it, there’s a bit of a difference there.” **Joe**: “When the team started to be at the top and everything became more intense, I felt that this wasn’t going to be as much fun, since the only thing was to win.” **Simon**: “There were some who wanted football as just a game, and those who wanted to put more into it, and that’s how it went.” **James**: “I knew that it wasn’t really what I wanted to do, but to have something to do, so I kept it going for years.”
	Growth of other interests and re-prioritising of time	**James**: “I think it was good to be able to decide for myself when I had to come up with something to do.” **Jim**: “I do not want to waste my life on football” **Simon**: “It all became a bit too much when I was working, refereeing as well as playing.” **Tom**: “I have more school now, and I’ve got an apprenticeship now, so I want to finish something, instead of just doing activities and football. I also found other interests that seemed a bit more interesting at that time, so I went in that direction instead.”
	Lack of physique and skills	**Didric**: “I came to puberty really late, and when all the others in the team and in the opposition got much bigger than me, and I became just little and kiddy in comparison.” **Didric**: “In the end, I couldn’t compete, because they were all bigger, stronger and speedier. I remember it being very frustrating.” **Joe**: “When the average level became so high that it was no fun anymore, then, in a way, it was all over.” **James:** “I started strength training alongside football because I felt too small and thin. The combination of these was a bit too heavy in the end.”
**Reasons for continued participation in football**	The social aspect of football	**Gary**: “I would have liked it to be like it was at the age of, when all my friends were together.” **Marcus**: “You could say that I lost a certain amount of motivation. That was the big carrot every season, and then it was taken away.” [talking about the biggest social event for the team each year] **Gary**: “There’s a big difference between running intervals with people you enjoy being with, and with people you don’t enjoy being with.” **Phil:** «If I hadn’t had quite a few friends who went to football, I probably wouldn’t have started it myself.”
	Getting fit through football	**Tom**: “It’s fun playing football. Free fitness training!” **Patric:** “Corona [and football break] showed how important football was for my physical fitness, because it was very difficult to start again after the break.” **Tim:** “I play football because I think everything about the sport is fun, and because it is good exercise.” **Marcus:** “After all, football is most of my training too, so when I’ve had a holiday, I notice that my performances are a bit worse at the start.”
	Finding it fun to play and compete	**Chris**: “I really love football” **Patric:** I play football now because I think it’s fun to play and show off, even if I don’t think I’ll be that good anymore.” **Tom:** “The team I play for now, is for those who just want to have fun with football, and who don’t want to become so good anymore. I’ve only played football all my life because I think it’s a lot of fun”. **Chris**: “I have a winning mentality, so I just have to win at something.”
	Ambition as a driving force	**Chris:** “For a period, I was so into football that I forgot everything else. I changed teams because I wanted to train as much as possible and become as good as possible.” **Marcus:** “My aim was to be taken out on the reginal top team, but when I didn’t, I felt my motivation dropped a little. It wasn’t as much fun to train anymore.” **Gary:** “I probably won’t go up to top leagues in the future, but I still have the goal of playing on a decent senior team when I get older. I want to have fun with football and go as far as possible.” **Patric:** “I trained a lot more before, when I had ambitions to play in the highest national league. Those goals have changed a bit now that I’m older and more realistic.”

## Data Availability

The raw data supporting the conclusions of this article will be made available by the authors, without undue reservation.

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
