# Peer review of "Dropping Out or Continuing Playing—A Case Study of Adolescent’s Motives for Participation in Football"

_sports, 2023, doi:10.3390/sports11070128_

Round 1

Reviewer 1 Report (Previous Reviewer 2)

The aim of this research is to find out the reasons why young people stop playing organised football or the reasons why they play football. The research is well thought out, well written, but lacks novelty. The manuscript does not state what is new about this research.

Do you not think that variables not discussed in the methodology, such as social class, may influence the motivations recorded?

The conclusions of this study cannot be generalized to all young people (as is done in the manuscript). In any case, it would be extrapolable to that country, but in other countries the football level, the probability of reaching the first division and other circumstances are different.

It is the editor's decision whether a well-written manuscript, with a correct methodology, but with a small sample and with very little novelty, has a place in his journal.

Author Response

Reviewer 1:

Author responses

The aim of this research is to find out the reasons why young people stop playing organised football or the reasons why they play football. The research is well thought out, well written, but lacks novelty. The manuscript does not state what is new about this research.

As the reviewer correctly pointed out the article lacked this in the first submission, but this was  included in second submission of the article: “Including in-depth data from all players at a football team – both players that dropped out and continued playing football, and also including some short interviews data from a longitudinal perspective, this study bring new findings into the discussion about sport participation.»

Do you not think that variables not discussed in the methodology, such as social class, may influence the motivations recorded?

It could be, but this was not examined. However, social class is suggested to be included in future research. The article is rewritten according to the reviewers comments

The conclusions of this study cannot be generalized to all young people (as is done in the manuscript). In any case, it would be extrapolable to that country, but in other countries the football level, the probability of reaching the first division and other circumstances are different.

I agree. As the reviewer correctly pointed out this was an error in the first submission, but this was  included in second submission of the article: ” One of the weaknesses of the study is that it is not possible to generalise the findings to all youth population from a small non-randomised sample. Furthermore, the study has not an experimental setting with a control group. However, the results may be valid for Norwegian adolescents, and for adolescents in other countries similar to Norway in relation to the organization of football.”

It is the editor's decision whether a well-written manuscript, with a correct methodology, but with a small sample and with very little novelty, has a place in his journal.

I will argue that the number of participants is high within this strategy of using in-depth interviews of all 13 players from a team, and analyzing these interviews focusing on meanings (as described by Johannessen, Tufte, and Kristoffersen [25]), and also that the novelty is satisfying

Reviewer 2 Report (Previous Reviewer 1)

In this study the authors aimed to investigate the motives leading to dropout among boys in youth football, as well as looking at motives why young people choose to continue with football.

Although the study seems to be interesting the manuscript is inconsistent in its methodology.

This is a serious aspect, which directly leads to the rejection of the manuscript.

The theoretical framework is scarce.

There is no experimental procedures

The Discussion should be enriched with the existing theory.

Author Response

Reviewer 2:

Author responses

Although the study seems to be interesting the manuscript is inconsistent in its methodology. This is a serious aspect, which directly leads to the rejection of the manuscript.

There was some inconsistent in the methodology in the first submission, but this was  included in second submission of the article. The methodology is well described and consistent within the tradition of qualitative designs, and in accordance with other published articles using  meaning condensation of in-depth interview data. The analyses were based on transcribed answers focusing on meanings, as described by Johannessen, Tufte, and Kristoffersen [25], which is a well-known scientific approach related to the use of interview data. However, there were some missing part in the first submission, that was  included in second submission of the article.

The theoretical framework is scarce.

The article is rewritten according to the reviewers comments

There is no experimental procedures

That is because this study is not an experimental study. The methodology is well described and consistent within the tradition of qualitative designs, and in accordance with other published articles using  meaning condensation of in-depth interview data.

The Discussion should be enriched with the existing theory.

The article is rewritten according to the reviewers comments

Reviewer 3 Report (New Reviewer)

Dear authors, first of all I would like to congratulate you for having carried out this very interesting research. 

After having proceeded to carry out a thorough reading, I miss the presence of the hypotheses of the research. Also, in the procedure section at the beginning, I would like to add the reason or reasons that led to the realization of this research. 

For the rest, congratulations once again for having carried out this research.

Author Response

Reviewer 3:

Author responses

Dear authors, first of all I would like to congratulate you for having carried out this very interesting research. 

Thank you

After having proceeded to carry out a thorough reading, I miss the presence of the hypotheses of the research.

The article is rewritten according to the reviewers comments

Also, in the procedure section at the beginning, I would like to add the reason or reasons that led to the realization of this research. 

The article is rewritten according to the reviewers comments

For the rest, congratulations once again for having carried out this research.

Thank you

Reviewer 4 Report (New Reviewer)

Dear Authors,

I would like to express my gratitude regarding the opportunity to review this manuscript.

At this stage the document requires considerable improvements:

It is suggested that lines are included in the document to allow for a more detailed review.

  Authors and affiliations are missing at the beginning of the document.   The template is from 2022, it should be updated.   On page 2 and other pages the letter font is different.   Some paragraphs have little text and others have a lot, please try to standardize to improve reading conditions.   There is no indication of the code of ethics associated with the study.   Page 4 paragraph ends without a end point and the next paragraph starts in lowercase, there must be errors here.   Page 5 - It is suggested to remove the name of the subjects and insert the number.   Results and discussion section seems close together and is confusing, revision suggested.   Conclusions should present paragraphs to improve reading conditions.   Missing: Supplementary Materials; Author Contributions; Funding; Institutional Review Board Statement; Informed Consent Statement; Data Availability Statement; Acknowledgments; Conflicts of Interest.   References must be reviewed in detail, they are not in accordance with the instructions for authors.   After considering the above points the document will be ready for the review process.            

Moderate editing of English language.

Author Response

Reviewer 4:

Author responses

I would like to express my gratitude regarding the opportunity to review this manuscript.

At this stage the document requires considerable improvements:

It is suggested that lines are included in the document to allow for a more detailed review.

The article is rewritten according to the reviewers comments

Authors and affiliations are missing at the beginning of the document.   The template is from 2022, it should be updated.   

This is an editorial responsibility that will be fixed by the editorial team before publishing. I have given this information

On page 2 and other pages the letter font is different.   

The article is rewritten according to the reviewers comments

Some paragraphs have little text and others have a lot, please try to standardize to improve reading conditions.  

The article is rewritten according to the reviewers comments

There is no indication of the code of ethics associated with the study.   

This is now included in the text: “The research project was approved by the Norwegian Centre for Research Data, fulfilling the ethical standards for empirical research. The respondents were fully informed about the protocol prior to participating in the study, and written consent was obtained from all. The study was conducted in accordance with the Declaration of Helsinki.

Page 4 paragraph ends without a end point and the next paragraph starts in lowercase, there must be errors here.   

The article is rewritten according to the reviewers comments

Page 5 - It is suggested to remove the name of the subjects and insert the number.   

The use of names is in consistence with the tradition of qualitative designs, and in accordance with other published articles using  meaning condensation of in-depth interview data. I think the strategy of using names makes the story  more personal.

Results and discussion section seems close together and is confusing, revision suggested.  

The use of a chapter where the results and discussion are put together, is names is in accordance with the tradition of qualitative designs, and in accordance with other published articles using meaning condensation of in-depth interview data.

Conclusions should present paragraphs to improve reading conditions.   

The article is rewritten according to the reviewers comments

Missing: Supplementary Materials; Author Contributions; Funding; Institutional Review Board Statement; Informed Consent Statement; Data Availability Statement; Acknowledgments; Conflicts of Interest.   

This is an editorial responsibility that will be fixed by the editorial team before publishing. I have given this information

References must be reviewed in detail, they are not in accordance with the instructions for authors.   After considering the above points the document will be ready for the review process.

The article is rewritten according to the reviewers comments

Reviewer 5 Report (New Reviewer)

It is an important topic. I hope my comment help.

Abstract, first sentence – why not investigate dropout and continuation motives?

Comma after 12) seems unnecessary.

Choppy abstract. Rewriting into shorter sentences will help and using the MS Word editor function.

Introduction

Perhaps a heading after your first paragraph 1.1. Sport Participation Benefits

2005 to 2011 statistics are way out of date.

1.2. Participation Factors

Or set up your first paragraph to indicate the section that will follow.

Good information. Using the MS Word editor function and working on the blue underline suggestions will help.

The dropout paragraphs seem unorganized in themes. Just a long list of reasons.

Given all the information you presented, how can you not have hypotheses to test? Or why is football so different than all the other sports in the world that your question is needed (without hypotheses)?

Materials and Methods

The authors being aware seems very biased from the start. Are any youth children of the authors or friends of friends, etc.?

The coaches mention ambition and then this turns into the main finding.

Reading

              the text

Again, the manuscript will benefit from a few hours of using the MS Word editor function.

Table 1

I do not see benefit to centering the sample quotations.

Your subheadings need to be set up, so they are obvious.

Discussion

Same comment a full clean with an editor function and perhaps you used one will help the reader/reviewer.

Knowing of the team seems a major weakness. The team selection is very biased and even personal.

I have not searched the literature. This cannot be the first study on this topic in football. It is a massive literature.

A full edit and explanation of the team is required.

Needs a full edit with MS Word or Grammarly or something like that. MS Word is easiest.

Author Response

Reviewer 5:

Author responses

Abstract, first sentence – why not investigate dropout and continuation motives?

The article is rewritten according to the reviewers comments

Comma after 12) seems unnecessary.

The article is rewritten according to the reviewers comments

Choppy abstract. Rewriting into shorter sentences will help and using the MS Word editor function.

The article is rewritten according to the reviewers comments, using the MS Word editor function

Introduction

Perhaps a heading after your first paragraph 1.1. Sport Participation Benefits

The article is rewritten according to the reviewers comments

2005 to 2011 statistics are way out of date.

I partly agree, but newer research is also included, and the research shows that much of the same factors appears

1.2. Participation Factors

Or set up your first paragraph to indicate the section that will follow.

The article is rewritten according to the reviewers comments

Good information. Using the MS Word editor function and working on the blue underline suggestions will help.

The article is rewritten according to the reviewers comments, using the MS Word editor function

The dropout paragraphs seem unorganized in themes. Just a long list of reasons.

The dropout paragraphs are organized according to different areas, as social relationship, structural limitations, and technical skills.

Given all the information you presented, how can you not have hypotheses to test? Or why is football so different than all the other sports in the world that your question is needed (without hypotheses)?

The article is rewritten according to the reviewers comments

Materials and Methods

The authors being aware seems very biased from the start. Are any youth children of the authors or friends of friends, etc.?

The first author who have done the interviews and the analyses of the data, are from another town and have no connection to the team. It is usual to use a stratified selection in interview studies, and the reflections of the participants probably reflects general Norwegian adolescents’ reflections. With such a strategy, the findings cannot be generalized – a fact that is highlighted in the study. However, the article is rewritten according to the reviewers comments, and the comment related to bias is included.

The coaches mention ambition and then this turns into the main finding.

The finding related to ambition takes place in the participants stories about their ambition, both in the in-depth interviews and the short interviews

Reading the text

Again, the manuscript will benefit from a few hours of using the MS Word editor function.

The article is rewritten according to the reviewers comments, using the MS Word editor function

Table 1

I do not see benefit to centering the sample quotations.

I agree. The article is rewritten according to the reviewers comments

Your subheadings need to be set up, so they are obvious.

The article is rewritten according to the reviewers comments

Discussion

Same comment a full clean with an editor function and perhaps you used one will help the reader/reviewer.

The article is rewritten according to the reviewers comments, using the MS Word editor function

Knowing of the team seems a major weakness. The team selection is very biased and even personal.

The article is rewritten according to the reviewers comments, and the comment related to bias is included.

I have not searched the literature. This cannot be the first study on this topic in football. It is a massive literature.

I have done a comprehensive literature search, and actually “This study is the first to examine data from all players at a football team, including both players that stopped and continued playing football”.

A full edit and explanation of the team is required.

The article lacked some information about the players in the first submission, but more was included in second submission of the article. Now the participants are described using 382 words.

Round 2

Reviewer 1 Report (Previous Reviewer 2)

We congratulate the authors because they have been able to respond to the contributions of the reviewers. Although the manuscript is adequately written, the editor should assess whether the research is of sufficient level and novelty to be published in his or her journal.

Author Response

Reviewer 1

Author response

We congratulate the authors because they have been able to respond to the contributions of the reviewers. Although the manuscript is adequately written, the editor should assess whether the research is of sufficient level and novelty to be published in his or her journal.

Thank you very much.

Reviewer 2 Report (Previous Reviewer 1)

No further comments

Author Response

Reviewer 2

Author response

No further comments

Thank you very much.

Reviewer 4 Report (New Reviewer)

Dear Authors,

Thank you for considering my suggestions and incorporating them into the manuscript, which globally improved, congratulations. Below some specific suggestions with line indication, but the manuscript at this point (v2) contains many errors in formatting and some in the text, which require a very careful analysis.

1,2- Title in the journal normally in uppercase. Please revise according to journal template and instructions for authors.

The authors responded regarding the authors and affiliations identification that “This is an editorial responsibility that will be fixed by the editorial team before publishing. I have given this information”. Please make sure of this.

35, 57 and other lines – The subtopics are not according to journal template and instructions for authors. Please revise.

107-109 – Text format is not according to the manuscript, journal template and instructions for authors. Please revise.

111-122 – Beginning of the paragraph space and space between paragraphs are not according to the journal template and instructions for authors. Please revise.

127 – Please revise the text, it is believed to be “In”, not “I”.

141 – N=13. Please address the sample power of the study.

163 – Please delete.

232-235 – Please delete.

236 – Two end points, please correct.

236 – “The use of names is in consistence with the tradition of qualitative designs, and in accordance with other published articles using meaning condensation of in-depth interview data. I think the strategy of using names makes the story more personal.” I do not agree with the author´s argumentation for using personal names, and I believe this may raise to ethical concerns. I once again suggest to replace names by “subject 1; subject 2;…”.

241 – The start of the paragraph format is not correct. Same in 619.

401 – Text format is incorrect.

518-524 – Text format is not according to journal template and instructions for authors. Please revise.

238-723 – Too many text, long paragraphs, makes it really hard to read a understand the rationale of the study, results, discussion and conclusions. Please consider changing the format in these sections.

724 – Journal template once again is not considered. It is missing “author´s contribution” and other information’s.

All references should be carefully revised; they are not according to the journal template and journal instructions for authors.

Please carefully revise the English throughout the manuscript and carry out a detailed analysis before v3 is concluded. As mentioned in the beginning, the manuscript at this point contains many errors in formatting and some in the text.

Moderate editing of English language required.

Author Response

Reviewer 4 comments

Author response

Thank you for considering my suggestions and incorporating them into the manuscript, which globally improved, congratulations. Below some specific suggestions with line indication, but the manuscript at this point (v2) contains many errors in formatting and some in the text, which require a very careful analysis.

Thank you very much.

1,2- Title in the journal normally in uppercase. Please revise according to journal template and instructions for authors.

The article is rewritten according to the reviewers comments

The authors responded regarding the authors and affiliations identification that “This is an editorial responsibility that will be fixed by the editorial team before publishing. I have given this information”. Please make sure of this.

Yes, I will make sure of this.

35, 57 and other lines – The subtopics are not according to journal template and instructions for authors. Please revise.

The article is rewritten according to the reviewers comments

107-109 – Text format is not according to the manuscript, journal template and instructions for authors. Please revise.

The article is rewritten according to the reviewers comments

111-122 – Beginning of the paragraph space and space between paragraphs are not according to the journal template and instructions for authors. Please revise.

The article is rewritten according to the reviewers comments

127 – Please revise the text, it is believed to be “In”, not “I”.

The article is rewritten according to the reviewers comments

141 – N=13. Please address the sample power of the study.

The sample power of the study is an important measure in quantitative studies, but are not addressed in qualitative studies, using long in-depth interviews from a low number of participants.

163 – Please delete.

The article is rewritten according to the reviewers comments

232-235 – Please delete.

The article is rewritten according to the reviewers comments

236 – Two end points, please correct.

The article is rewritten according to the reviewers comments

236 – “The use of names is in consistence with the tradition of qualitative designs, and in accordance with other published articles using meaning condensation of in-depth interview data. I think the strategy of using names makes the story more personal.” I do not agree with the author´s argumentation for using personal names, and I believe this may raise to ethical concerns. I once again suggest to replace names by “subject 1; subject 2;…”.

I agree with you that the strategy of using names makes the story more personal. I totally agree with you according to the ethical concerns related to using correct names. However, the participants have been given pseudonyms in the presentation of the analysis. 

This is now highlighted in the text, and I hope this is fine.

241 – The start of the paragraph format is not correct. Same in 619.

The article is rewritten according to the reviewers comments

401 – Text format is incorrect.

The article is rewritten according to the reviewers comments

518-524 – Text format is not according to journal template and instructions for authors. Please revise.

The article is rewritten according to the reviewers comments

238-723 – Too many text, long paragraphs, makes it really hard to read a understand the rationale of the study, results, discussion and conclusions. Please consider changing the format in these sections.

The article is rewritten according to the reviewers comments

724 – Journal template once again is not considered. It is missing “author´s contribution” and other information’s.

This information has been given, and I will make sure that they will be included

All references should be carefully revised; they are not according to the journal template and journal instructions for authors.

The article is rewritten according to the reviewers comments. Errors here will also be detected in the proof version

Please carefully revise the English throughout the manuscript and carry out a detailed analysis before v3 is concluded. As mentioned in the beginning, the manuscript at this point contains many errors in formatting and some in the text.

The article is rewritten according to the reviewers comments

Comments on the Quality of English Language

Moderate editing of English language required.

The article is rewritten according to the reviewers comments

Reviewer 5 Report (New Reviewer)

Hello, I appreciate your revision efforts. I did not intend for any conflict or tension if perceived. Perhaps your work is the first. I was thinking of sport in general. Here is one reference. 

https://www.mdpi.com/2075-4663/6/4/165

My point is not that you need to reference the manuscript. These are quantitative data, not qualitative. There is a a time 1 and time 2 (dropouts and those who continued). That was my point data are around with dropouts and non-dropouts.

Minor corrections and formatting is required.

Author Response

Reviewer 5

Author response

Hello, I appreciate your revision efforts. I did not intend for any conflict or tension if perceived. Perhaps your work is the first. I was thinking of sport in general. Here is one reference. https://www.mdpi.com/2075-4663/6/4/165

Thank you. I agree with your comments related to sports in general – other studies has looked upon drop out or reaching elite/non elite level (as your reference exemplifies), but it is the first to look at a complete team, interviewing both drop out/not drop out players in time 1 and 2.

The reference is included in the article

My point is not that you need to reference the manuscript. These are quantitative data, not qualitative. There is a a time 1 and time 2 (dropouts and those who continued). That was my point data are around with dropouts and non-dropouts.

I agree. There is a time 1 and time 2. My point was only that the data is short interviews and in-depth interviews (words, not numbers), and therefore qualitative data. Hopefully we have the same reflections.

This manuscript is a resubmission of an earlier submission. The following is a list of the peer review reports and author responses from that submission.

Round 1

Reviewer 1 Report

In this study the authors aimed to investigate the motives leading to dropout among boys in youth football, as well as looking at motives why young people choose to continue with football.

Although the study seems to be interesting the manuscript is inconsistent in its methodology which does not support the experimental setting.

This is a serious aspect, which directly leads to the rejection of the manuscript.

1.      Moreover, the theoretical framework is scarce, they should clearly describe the scientific evidence that supports the hypothesis they have raised.

1.               A lot of necessary information is missing in methods section:

-        Experimental procedures should be better defined

-        More information should be provided about the participants’ characteristics.

-        The intervention protocol should be better described.

This element is missing from the methodological description, which may imply an impossibility of replicating the study due to a lack of clarity in this regard.

2.               The Discussion should be enriched with the existing theory. The authors should clearly describe the scientific evidence that supports their findings. In addition, they should start with a first paragraph describing the main aims and then the main results.

Reviewer 2 Report

The aim of this research is to find out the reasons why young people stop playing organised football or the reasons why they play football. The research is well thought out, well written, but lacks novelty. The manuscript does not state what is new about this research.

Do you not think that variables not discussed in the methodology, such as social class, may influence the motivations recorded?

The conclusions of this study cannot be generalized to all young people (as is done in the manuscript). In any case, it would be extrapolable to that country, but in other countries the football level, the probability of reaching the first division and other circumstances are different.